# M²oT: Agglomerative Vision Foundation Models via Sparse Mixture-of-Experts

## Abstract

Agglomerative models aim to unify the strengths of various vision foundation models through multi-teacher distillation for enhanced performance across diverse tasks. However, current feature-aligned distillation approaches for agglomerative models frequently encounter a compromised trap: student models learn compromised features that overlook the unique contributions and inherent differences of individual teachers, limiting the overall performance of models. To mitigate this limitation, we propose a novel Sparse Mixture-of-Experts (SMoE) based framework for Multi-Teacher distillation (M²oT). Within M²oT, we introduce a teacher-aware loss as a regularization term to actively increase expert diversity, enabling the SMoE to capture specialized features tailored to each teacher's unique contributions. Extensive experiments have demonstrated the superior performance of our method across various large-scale vision tasks, validating its effectiveness in resolving the compromised trap and enhancing overall model performance.

## 1 Introduction

Recent studies focus on agglomerative foundation models (Heinrich et al., 2025), combining strengths from multiple Vision Foundation Models (VFMs). Popular VFMs like DINOv2 (Oquab et al., 2024), CLIP (Radford et al., 2021), and SAM2 (Ravi et al., 2024) show strong performance on various downstream tasks. To transfer this knowledge, AM-RADIO (Ranzinger et al., 2024) distilled knowledge from pre-trained models without ground truth. This multi-teacher framework proved highly effective, enabling smaller student networks to internalize collective teacher expertise, fostering robust vision systems.

However, current methods aim for comprehensive knowledge transfer by aligning student features with each teacher. However, they face a significant inherent limitation we term the **compromised trap**. Even with techniques like teacher-specific projection heads (Sarıyıldız et al., 2024; 2025), the student's shared feature extractor learns a generalized representation, resulting in compromised features. This occurs because the extractor must simultaneously encode sufficient information for all teachers, frequently obscuring unique, fine-grained, specialized knowledge from individual teachers and limiting the student's full potential.

To address this and boost specialized knowledge integration, Sparse Mixture-of-Experts (SMoE) architectures offer a compelling solution. MoE models excel at processing diverse inputs by routing them to specialized sub-networks (experts). This massively increases model capacity and enables specialized learning without proportional computational cost. Despite their advantages, SMoE's full potential in multi-teacher agglomerative VFM distillation remains largely unexplored. Effectively leveraging MoE's modularity to genuinely integrate heterogeneous teacher knowledge, rather than merely averaging it, is a critical unmet need.

In this paper, we propose a novel framework. It combines SMoE with an innovative distillation strategy to overcome the compromised trap and achieve superior agglomerative models. Our framework introduces the SMoE architecture into multi-teacher agglomerative VFMs, providing needed capacity and modularity for specialized learning. We design a novel teacher-aware loss function. It actively guides SMoE experts to learn context-aware, teacher-specific specializations. This loss ensures each expert prioritizes alignment with relevant teacher knowledge, directly combating dilution

and fostering experts capable of capturing truly unique and valuable specialized features. Figure 1 shows our method's framework.

In summary, our contributions are threefold:

- We pioneer the use of Sparse Mixture-of-Experts architectures in multi-teacher agglomerative vision foundation models, providing a scalable, modular backbone for consolidating diverse teacher knowledge.

- We propose a novel teacher-aware loss function. It significantly increases the expert diversity of SMoE and guides experts toward teacher-specific specialization. This loss approximates a Bayesian Maximum A Posteriori inference objective for expert-specific teacher targets.

- We empirically show our framework solves the compromised trap. The student model moves beyond learning compromised features to capture more discriminative and task-relevant specialized features. This is evidenced by an increase in conventional distillation loss but significant performance gains across various vision tasks.

## 2 RELATED WORKS

### 2.1 AGGLOMERATIVE VISION FOUNDATION MODELS

**Unified models.** Combining and unifying the capabilities of multiple models from various vision tasks and heterogeneous domains is the central goal of agglomerative models in computer vision. To build such agglomerative models, related strategies such as model merging (Zhang et al., 2024), assuming that models have the same architecture and size, multi-task learning (Lu et al., 2025), which requires expensive labels and introduces task-specific bias, and continual learning (Chen et al., 2023), which faces the challenge of catastrophic forgetting for previous tasks.

**Multi-teacher distillation.** Applying multi-teacher knowledge distillation is another way to build agglomerative models. It constructs a single student vision encoder distilled from multiple vision teachers. AM-RADIO (Ranzinger et al., 2024) proposed to match the features of teachers without using labels to cover the heterogeneous domains, which is a dominant multi-teacher distillation framework for agglomerative models. Theia (Shang et al., 2024) applied this paradigm to establish a unified model for robotic applications with a selected set of teachers. UNIC (Sarıyıldız et al., 2024) proposed to join feature standardization to balance teachers and teacher-dropping regularization to preserve the accuracy of top teachers, especially for classification tasks. RADIOv2.5 (Heinrich et al., 2025) extended their focus on the robustness to various image resolutions through multi-resolution training. DUNE (Sarıyıldız et al., 2025) studied special 3D teachers for heterogeneous domains while employing attention-based projectors for teacher-specific datasets.

In this paper, we argue that previous multi-teacher studies for agglomerative VFMs have suffered from seeking compromised features among teachers, limiting to obtain the unique features of each teacher.

### 2.2 MIXTURE-OF-EXPERTS

**Sparse Mixture-of-Experts.** Current transformer architecture (Vaswani, 2017) is commonly adopted in large models such as large language models (Achiam et al., 2023) and diffusion models (Peebles & Xie, 2023). Sparse Mixture-of-experts (SMoE) models are proposed to address the scaling computing demand, which sends different tokens of an input sequence to different experts (Riquelme et al., 2021; Zhu et al., 2024a; Dai et al., 2024). Ongoing efforts in SMoE research largely emphasize load balancing (Fedus et al., 2022) to prevent experts from becoming underutilized or overloaded.

**Expert Specialization and Diversity.** A key challenge in SMoE models is to ensure that experts truly learn distinct representations and that their collective capacity is effectively utilized. Previous studies (Riquelme et al., 2021; Chen et al., 2022) have verified the existence of expert specialization for SMoE in various domains. Recent works (Qiu et al., 2025; Yuan et al., 2025) analyze that promoting expert specialization is a trade-off to load balancing. To promote expert specialization and

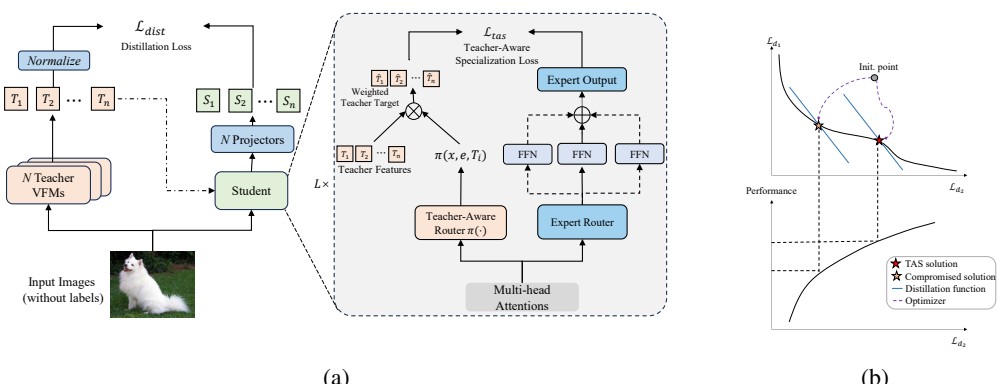

(a) (b)

Figure 1: Overview of the $M^2$oT framework. (a) (*Left.*) illustrates the multi-teacher distillation setup, highlighting the distillation loss, and (*Right.*) details the SMoE student architecture and explains how $\mathcal{L}_{\text{tas}}$ is computed to enhance expert diversity. (b) depicts the advantage of our method's solution over a compromised solution from a Pareto Frontier perspective (*Top.*) and in terms of downstream performance (*Bottom.*) for a two-teacher scenario, denoting the individual teacher losses as $\mathcal{L}_{d_1}$ and $\mathcal{L}_{d_2}$.

diversity, Upcycling-based method (Zhu et al., 2024b; Nakamura et al., 2025) employs diverse dense checkpoints to initialize different experts, yet questions arise if the initialization of experts affects expert diversity as a result of training processes. In this paper, we introduce a regularization loss with the prior of teacher features to increase expert diversity and aim to improve expert specialization for SMoE models.

## 3 PRELIMINARIES AND ANALYSIS

### 3.1 MULTI-TEACHER DISTILLATION FRAMEWORK

In this paper, our multi-teacher distillation framework for agglomerative models mainly follows (Ranzinger et al., 2024). Specifically, we build a student visual encoder as a foundation model based on the Transformer architecture like ViT (Dosovitskiy et al., 2021). Let image $x \in \mathcal{I}$ be the input and $z = f(x), z \in \mathcal{Z}$ be the feature vector produced by the visual encoder, where $\mathcal{Z} \in \mathbb{R}^{(S+1) \times d}$ and $f(\cdot)$ denotes the student encoder network. Each feature vector produced in a certain network layer has a dimension of $d$. The feature set consists of two kinds of tokens: 1) $S$ patch tokens and 2) an optional CLS token, which represents the global information about an image.

We distill our student model by aligning the features between teachers and the student model. The teacher set $T = \{T_1, ..., T_N\}$ denotes $N$ teacher encoders. The student model shares the same backbone for all teachers. For better alignment of the features, adapted heads are applied to project the shared student features to the specific teacher feature space, named projectors (Sarıyıldız et al., 2025). In other words, for a teacher $T_i$ indexed by $i$, the projectors transform shared student features $z$ to teacher-specific features $\hat{z}_i$, as defined in Eq. 1.

$$\hat{z}_i = g_i(z), \; g_i(z) : \mathbb{R}^d \to \mathbb{R}^{d_i} \tag{1}$$

where $g(\cdot)$ is commonly implemented as multi-layer perceptrons (MLPs) and $d_i$ is the dimension of feature vectors produced by the teacher $T_i$. As two kinds of features are employed, we define two corresponding projectors for both patch tokens and CLS tokens in each network layer.

For the distillation loss under the setting of feature distillation, we combine both cosine-similarity and smooth-$l_1$ losses to measure the distance for features between multiple teachers and the student, and minimize them as our distillation loss function among all teachers, formulated as Eq. 2.

$$\mathcal{L}_{\text{dist}} = \sum_{i=1}^{N} \mathcal{L}_{cos}(h_i(x), T_i(x)) + \mathcal{L}_{sl_1}(h_i(x), T_i(x)) \tag{2}$$

where $h_i := f \cdot g_i$. Both distance metrics are defined in (Sarıyıldız et al., 2024).

## 3.2 COMPROMISED TRAP OF DISTILLATION LOSS

Agglomerative multi-teacher distillation often leverages a distillation loss to align student features with individual teachers. We refine the loss function in Eq. 2 into a general form, formulated as Eq. 3.

$$\mathcal{L}_{\text{dist}} = \mathbb{E}_{x \sim D}\left[\sum_{i=1}^{N} \text{Dist}(h_i(x), T_i(x))\right] \tag{3}$$

where $D$ represents the datasets and $\text{Dist}(\cdot, \cdot)$ is a chosen distance metric (e.g., smooth-$l_1$, cosine-similarity, mean-square error).

**Proposition 1 (Compromised Trap)** *Let $f_i^* = \arg\min_f \mathbb{E}_{x \sim D}\left[Dist(f, T_i; x)\right]$ be the optimal encoder on a single teacher. With $\hat{f} = \arg\min_f \mathcal{L}_{dist}(f; x)$ as the encoder under multiple teachers, there exists teacher $T_i$ that $\mathcal{L}_{dist}(\hat{f}, T_i) > \mathcal{L}_{dist}(f_i^*, T_i)$ leading to limited performance in a certain task, i.e., $U(\hat{f}(x)) < U(f_i^*(x))$, where $U$ is a target utility function.*

As formalized in Proposition 1 (proof in Appendix A.5.2) and seen in Figure 2, the compromised trap arises in multi-teacher distillation. Figure 2 shows this: distillation loss against individual teachers is consistently higher in the multi-teacher setting than in the single-teacher setting. This happens because the shared student encoder $f(x)$ must simultaneously represent diverse or conflicting teacher information. Thus, $f(x)$ becomes a generalized, averaged representation, resulting in compromised features. This compromise, while minimizing overall $\mathcal{L}_{\text{dist}}$, sacrifices unique, fine-grained teacher details. This limits the student's ability to specialize or capture nuanced distinctions, leading to reduced downstream performance.

## 4 METHOD

We introduce M²oT, a novel framework for solving the compromised trap via Sparse Mixture-of-Experts. Our framework consists of two main components: 1) Sparse Upcycling Architecture of Dense Student: We design a sparse student network with MoE, enhancing the ability to acquire specialized teacher knowledge while maintaining training and inference efficiency. 2) Specialization-Oriented Knowledge Distillation Mechanism: We design a novel Teacher-Aware Specialization Loss, based on Maximum A Posteriori estimation, designed to mitigate this trap by guiding expert specialization and diversity.

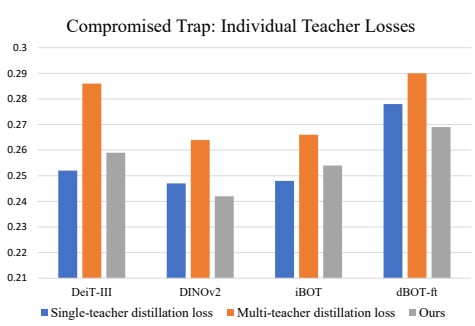

Figure 2: Comparison of distillation losses under single-teacher, multi-teacher distillation settings and our method, illustrating the compromised trap our mitigation on it.

## 4.1 SPARSE UPCYCLING ARCHITECTURE OF DENSE STUDENT

A Sparse Mixture-of-Experts (SMoE) layer replaces a standard Feed-Forward Network (FFN) in a transformer (denoted as a dense model) with a collection of expert FFNs. For an SMoE layer with $M$ experts, given an input of $n$ tokens $t \in \mathbb{R}^{n \times d}$ from the Multi-Head Attention (MHA) module, the output is a weighted sum of each expert's computation $e_j(t)$. This is determined by a routing function $R(t) \in [0, 1]$, as defined in Eq. 4.

$$f_{\text{SMoE}}(t) = \sum_{j=1}^{M} R(t)_j \cdot e_j(t) \tag{4}$$

where the routing function $R(t)$ employs $\text{TOP}_k$ operation to active a subset of $k$ experts, as formulated in Eq. 5.

$$R(t) = \text{TOP}_k(\text{softmax}(r(t))) \tag{5}$$

where $r(t) : \mathbb{R}^{n \times d} \to \mathbb{R}^{n \times M}$ is a router network, often adopted as an MLP. To reduce the overhead for training SMoE, we apply the sparse upcycling technique (Komatsuzaki et al., 2022) to replicate $M$ FFNs as expert modules from the dense student.

### 4.2 SPECIALIZATION-ORIENTED KNOWLEDGE DISTILLATION

**Teacher-Aware Specialization Loss.** To overcome the compromised trap, we introduce a novel Teacher-Aware Specialization (TAS) loss based on Sparse Mixture-of-Experts architectures. Our approach shifts from generic alignment to adaptive, expert-specific knowledge integration, grounded in Maximum A Posteriori (MAP) estimation. MAP estimation offers a principled framework for inferring ideal latent teacher targets from multiple observations.

We frame the problem as inferring an ideal, latent teacher target for each activated SMoE expert. Let $\mathcal{K}$ denote the set of top $k$ experts. For an activated expert $e \in \mathcal{K}$, we hypothesize its output $\phi(x, e)$ should align with an ideal target $\hat{T}^*(x, e)$. We consider the features from the $N$ teachers $\{T_i(x)\}_{i=1}^N$ as observations of this latent target. Assuming Laplace noise ($l_1$ distance) and a flat prior on $\hat{T}^*(x, e)$, the MAP estimate for $\hat{T}^*(x, e)$ minimizes the negative log-likelihood, as in Eq. 6.

$$\hat{T}^*(x, e) = \arg\min_{\hat{T}} \sum_{i=1}^{N} w_i \left\| T_i(x) - \hat{T} \right\|_1 \tag{6}$$

This expression defines $\hat{T}^*(x, e)$ as the weighted geometric median of teacher features $T_i(x)$, based on their relevance $w_i$. An ideal teacher-aware loss would guide expert $\hat{T}^*(x, e)$ towards this intractable target in Eq. 7.

$$\mathcal{L}_{\text{ideal}} = \mathbb{E}_{x \sim D} \left[ \sum_{e \in \mathcal{K}} \text{softmax}(r(x)) \cdot \left\| \phi(x, e) - \hat{T}^*(x, e) \right\|_1 \right] \tag{7}$$

However, computing the geometric median is prohibitive during training, and true relevance weights $w_i$ are unknown. To make this objective tractable, we use two approximations: 1) We approximate the weighted geometric median with the more efficient weighted arithmetic mean: $\hat{T}^*(x, e) \approx \sum_{i=1}^{N} w_i \cdot T_i(x)$, and 2) we replace unknown $w_i$ with outputs from our learnable teacher-aware router, $\pi(x, e, T_i)$, which dynamically estimates each teacher's relevance to expert $e$ for input $x$.

Applying these approximations, we derive our practical Teacher-Aware Specialization loss. It softly guides SMoE experts towards teacher-aware specialization and encourages activated experts to prioritize alignment with relevant teachers identified by the teacher selection module, as formulated in Eq. 8.

$$\mathcal{L}_{\text{tas}} = \mathbb{E}_{x \sim D} \left[ \sum_{e \in \mathcal{K}} \text{softmax}(r(x)) \cdot \left\| \phi(x, e) - \overline{T}(x, e) \right\|_1 \right] \tag{8}$$

where $\overline{T}(x, e) = \sum_{i=1}^{N} \pi(x, e, T_i) \cdot T_i(x)$. This formulation enables each SMoE expert to adaptively learn and align with a dynamically weighted combination of teacher features, effectively guiding them towards specialized features without falling into the compromised trap. The learnable teacher-aware router $\pi(\cdot)$ allows the model to infer and prioritize the most relevant teacher knowledge for each expert and input.

Table 1: Comparison of different distillation methods evaluated on ImageNet100 (IN100) and ADE20K datasets. We report the average values of the distillation loss ($\mathcal{L}_{\mathrm{dist}}$) and our proposed TAS loss ($\mathcal{L}_{\mathrm{tas}}$) during training.

| | $\mathcal{L}_{\mathrm{dist}}$ | $\mathcal{L}_{\mathrm{tas}}$ | IN100 | ADE20K |
|---|---|---|---|---|
| Dense | 0.187 | - | 92.50±0.06 | 35.18±0.14 |
| Sparse MoE | 0.145 | 3.27 | 93.40±0.07 | 37.46±0.31 |
| M²oT(Ours) | 0.156 | 1.45 | **93.84±0.04** | **37.91±0.19** |

From a computational view, a learnable teacher-aware router avoids computing all experts' outputs to infer the latent teacher target. This saves memory and computation for teacher-specific representations.

Our TAS Loss serves as a regularizer for the main distillation objective $\mathcal{L}_{\mathrm{dist}}$. Proposition 2 proves a distinct tradeoff between the TAS loss and the main distillation loss (proof in Appendix A.5.3).

**Proposition 2 (Expert Diversity as Regularizer)** *Let $f_1^*$ be the optimal student encoder obtained solely by minimizing $\mathcal{L}_{dist}$, and $f_2^*$ be the student encoder optimized with $\mathcal{L}_{tas}$ as a regularizer. There exists a scenario where, even if $\mathcal{L}_{dist}(f_2^*) > \mathcal{L}_{dist}(f_1^*)$, this leads to improved downstream task performance, i.e., $U(f_2^*(x)) > U(f_1^*(x))$, where $U$ is a target utility function.*

Low agglomerative distillation loss $\mathcal{L}_{\mathrm{dist}}$ does not correlate with high downstream performance. Table 1 compares distillation methods on ImageNet100 (Tian et al., 2020). Sparse MoE, with increased capacity, shows lower $\mathcal{L}_{\mathrm{dist}}$ and better downstream performance than Dense. Our M²oT method significantly reduces $\mathcal{L}_{\mathrm{tas}}$, indicating successful expert specialization. This specialization slightly increases $\mathcal{L}_{\mathrm{dist}}$. Despite this, M²oT delivers the highest performance on both ImageNet100 and ADE20K. This validates Proposition 2: strategically increasing distillation loss, driven by specialized feature integration, yields superior overall utility.

**Load-Balancing Loss.** SMoE models often suffer from the issue of imbalanced expert load. The router frequently tends to route tokens to a subset of experts within an SMoE layer, leading to the underutilization of other experts during training. This results in inefficient parameter and computation utilization. A load-balancing loss (Fedus et al., 2022) serves as an auxiliary loss function to ensure that each expert receives a comparable number of tokens within a batch, as defined in Eq. 9.

$$\mathcal{L}_{\mathrm{lb}} = M \sum_{j=1}^{M} F_j \cdot P_j \tag{9}$$

where $F_j$ denotes the fraction of tokens routed to each expert $e_j$ and $P_j$ denotes the total routing probability allocated to the expert $e_j$.

**Overall Loss.** In conclusion, the total loss function for feature-aligned distillation to update the student model parameter is defined as Eq. 10.

$$\mathcal{L}_{\mathrm{total}} = \mathcal{L}_{\mathrm{dist}} + \lambda_1 \mathcal{L}_{\mathrm{tas}} + \lambda_2 \mathcal{L}_{\mathrm{lb}} \tag{10}$$

The overall loss function combines three parts: 1) a multi-teacher distillation term, which aims to capture a comprehensive feature representation but can lead to compromised features within the student agglomerative model, 2) an expert specialization term, acting as a regularizer to promote the extraction of teacher-specific specialized features by increasing expert diversity, and 3) a load balancing term for SMoE, encouraging the utilization of experts.

## 5 EXPERIMENTS

### 5.1 EXPERIMENTAL SETTINGS

**Evaluation Benchmarks.** We evaluate our models across various vision tasks. For Image Classification, we report ImageNet1K (Russakovsky et al., 2015) k-NN classification using shared backbone

Table 2: State-of-the-art comparison of multi-teacher distillation agglomeration models. Performance is evaluated across ImageNet1K (IN1K) classification accuracy (↑), ADE20K semantic segmentation mIoU (↑), and NYU depth estimation RMSE (↓). Best results are in bold, second-best are underlined.

| Model | Active Params. | Encoder Arch. | Training Data | IN1K(↑) | ADE20K(↑) | NYUd(↓) |
|---|---|---|---|---|---|---|
| *Teacher Models* | | | | | | |
| DINOv2 | 86M | ViT-Base | LVD-142M | 82.01 | 41.10 | 0.481 |
| SAM2 | 224M | Hiera-Large | SA-V | - | 29.17 | - |
| *Multi-teacher Distillation* | | | | | | |
| SAM-CLIP | 86M | ViT-Base | ImageNet1K | - | 38.4 | - |
| Theia | 86M | ViT-Base | ImageNet1K | 81.19 | 35.55 | 0.637 |
| UNIC-B | 86M | ViT-Base | ImageNet1K | 83.21 | 39.60 | 0.547 |
| RADIOv2.5-B | 98M | ViT-Base | DataComp-1B | 81.96 | **48.94** | 0.498 |
| M$^2$oT-B(Ours) | 86M | ViT-Base | ImageNet1K | **84.28** | 42.50 | **0.479** |
| UNIC-L | 307M | ViT-Large | ImageNet1K | 85.60 | 48.30 | 0.491 |
| RADIOv2.5-L | 320M | ViT-Large | DataComp-1B | 84.68 | 51.47 | 0.457 |
| M$^2$oT-L(Ours) | 213M | Hiera-Large | ImageNet21K | **86.53** | **52.14** | **0.442** |

embeddings. For Semantic Segmentation, we assess models on ADE20K (Zhou et al., 2019) (mIoU) with an MMSeg (Contributors, 2020) framework, training a decoder on top of frozen features. For depth estimation, we evaluate on NYUdv2 (Silberman et al., 2012) (RMSE). We use a linear probe for ViT and UPerNet (Xiao et al., 2018) for multi-stage encoders like Swin Transformer (Liu et al., 2021). For Depth Estimation, we follow DINOv2 (Oquab et al., 2024) settings, building DPT (Ranftl et al., 2021) over the features.

**Implementation Details.** We use ImageNet1K and ImageNet21K (Ridnik et al., 2021) as distillation datasets, using only images for feature alignment. Our encoder architectures include ViT and a modified Hiera (Ryali et al., 2023), detailed in Appendix A.3.2.

Our Sparse MoE architecture is implemented via the Tutel (Hwang et al., 2023) framework. We use TOP$_2$ ($k = 2$) for most experiments, activating 2 experts per token out of 8 total experts. Every odd layer (e.g., 1, 3, ..., 11) is an MoE layer, and we incorporate our TAS loss in these layers as a sum. Load-balancing loss is applied with $\lambda_2 = 0.01$. Inspired by Fine-grained MoE (Ludziejewski et al., 2024), we reduce FFN hidden dimensions as $k$ increases, maintaining active parameters and FLOPs for fair comparison. In ablation studies, we omit sparse upcycling to isolate TAS loss performance.

**Teacher Models.** To distill sufficient unique features for multiple teachers as much as possible, we select a list of VFMs, including DeiT-III (Touvron et al., 2022), DINOv2 (Oquab et al., 2024), iBOT (Zhou et al., 2022), dBOT-ft (xingbin liu et al., 2024), SigLIP (Zhai et al., 2023), SAM2 (Ravi et al., 2025) and AIMv2 (Fini et al., 2025). These teachers contain heterogeneous knowledge in various training schemes and datasets.

## 5.2 COMPARATIVE RESULTS

Table 2 compares our method with other agglomerative VFMs: SAM-CLIP (Wang et al., 2024), Theia (Shang et al., 2024), UNIC (Sarıyıldız et al., 2024), and RADIOv2.5 (Heinrich et al., 2025). We evaluate base (B) and large (L) student models across various vision tasks, including two typical teacher models for comparison.

At the base scale, our M$^2$oT-B model, using a standard ViT-Base encoder and trained only on ImageNet1K, achieves highly competitive results. It outperforms all baselines on ImageNet1K accuracy and sets a new state-of-the-art on NYU depth estimation. While RADIOv2.5-B shows higher mIoU on ADE20K, it uses a much larger DataComp-1B dataset and slightly more active parameters. Our method still achieves a strong performance on ADE20K, showing robust performance with less training data.

For large-scale models, our M$^2$oT-L model, using a Hiera-Large encoder and trained on ImageNet21K, significantly advances performance. It decisively outperforms all large-scale baselines, achieving top scores across all three challenging tasks. Notably, M$^2$oT outperforms models like RADIOv2.5-L, which also use the massive DataComp-1B (Gadre et al., 2023) dataset. This highlights our specialized knowledge aggregation framework's effectiveness, scalability, and generaliz-

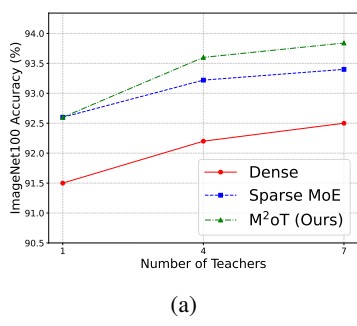 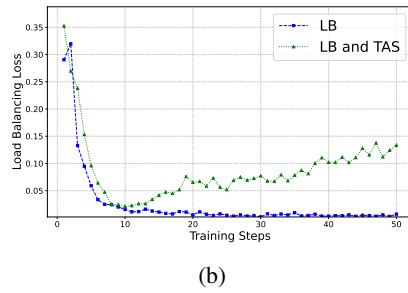

(a) (b)

Figure 3: (a) ImageNet100 accuracy across varying teacher numbers. (b) Load-balancing loss curve during training.

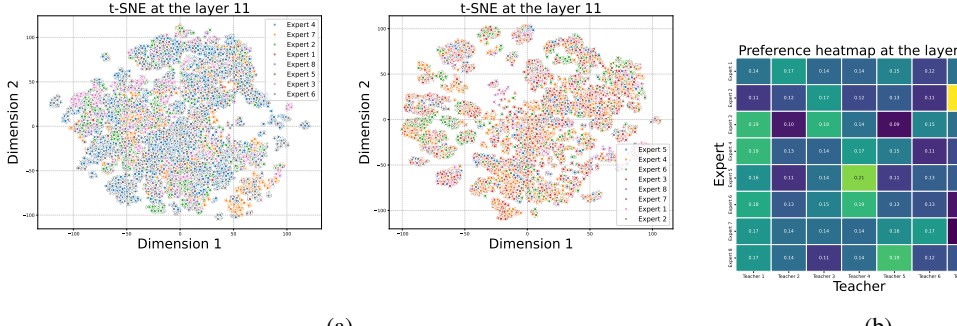

(a) (b)

Figure 4: Visualization of expert specialization and diversity. (a) t-SNE plots of expert features in SMoE layer without (*Left.*) and with (*Right.*) the TAS loss. (b) Expert-Teacher preference heatmap showing expert scores across different teachers.

ability, even against models trained on vast datasets and different encoders. Our method supports arbitrary multi-teacher distillation across various encoder architectures between teacher and student models, showing superior performance in both accuracy and computational efficiency.

## 5.3 ABLATION STUDIES

**Scaling the Number of Teachers.** Figure 3a illustrates the impact of scaling teacher number on ImageNet100 accuracy. All methods exhibit performance gains with more teachers, underscoring the potential of diverse knowledge. Despite inherent teacher heterogeneity, scaling the number of teachers is crucial to validate a method's ability to leverage this diversity and overcome the compromised trap. SMoE consistently outperforms Dense, confirming the effectiveness of SMoE, and our $M^2$oT method consistently achieves the highest accuracy across all teacher numbers. This demonstrates that TAS loss captures unique teacher features and the efficacy in exploiting rich, diverse teacher ensembles.

**Expert Specialization and Diversity.** Our analysis reveals how the TAS loss effectively addresses the compromise trap. We draw t-SNE (Cai & Ma, 2022) visualizations of expert features by 12608 patch features in the final (11th) MoE layer with 32 samples of ImageNet100 validation datasets, as shown in Figure 4a. Without the TAS loss, expert features are broadly scattered, reflecting compromised features that result from averaging diverse teacher knowledge. In contrast, with TAS loss, expert features become tightly clustered. This illustrates experts focusing on specialized features.

Further, the Expert-Teacher preference heatmap, as shown in Figure 4b, demonstrates that our $M^2$oT, unlike vanilla SMoE, promotes experts gaining significantly higher routing scores for specific teachers. This directly indicates the successful development of specialized features, where individual experts learn to prioritize and process unique information from particular teachers.

Table 3: Impact of the loss coefficient $\lambda_1$ for ImageNet100, ADE20K and NYUd datasets.

| $\lambda_1$ | IN100($\uparrow$) | ADE20K($\uparrow$) | NYUd($\downarrow$) |
|---|---|---|---|
| 0.01 | 93.38 | 37.10 | 0.601 |
| 0.10 | 93.67 | 37.47 | 0.592 |
| 0.50 | **93.84** | **37.91** | **0.585** |
| 1.00 | 93.42 | 37.41 | 0.597 |
| 5.00 | 92.12 | 35.25 | 0.623 |

Table 4: Comparison of memory usage and computation budget in training time.

| | Params. | Memory | GFLOPs | Time |
|---|---|---|---|---|
| Dense MoE (Oracle) | 170M | 74.2 | 81.19 | 3.47 |
| Dense | 86M | 37.9 | 35.15 | 1.49 |
| Sparse MoE | 170M | 39.1 | 35.34 | 1.51 |
| $M^2$oT(Ours) | 171M | 39.6 | 35.44 | 1.59 |

**Choice of Loss Coefficient.** Table 3 presents the impact of varying the coefficient $\lambda_1$ for our TAS loss on downstream task performance. We observe that setting $\lambda_1 = 0.5$ yields the optimal results, achieving the highest model performance. This indicates that $\lambda_1$ plays a crucial role in balancing the emphasis on expert specialization against the distillation objective.

**Load-Balancing Analysis.** In general, it is a trade-off between expert specialization and load balance for sparse MoE. To further identify the mechanism of Teacher-Aware Specialization Loss, we show the load-balanced loss in training. Figure 3b illustrates that our $M^2$oT method, incorporating the TAS loss, consistently exhibits a higher load balancing loss compared to the vanilla SMoE baseline. The elevated load-balancing loss for $M^2$oT is not a drawback, but rather a sign that the model effectively leverages its MoE structure to assign inputs to the most competent and specialized experts, avoiding overbalancing for SMoE layers (Qiu et al., 2025).

**System Resource Analysis.** As we add an extra router for computing the TAS term during training time, we collect total model parameters, memory footprints (GB), computational footprints (GFLOPs), and training time (min/step) of different student model configurations shown in Table 4. The Dense MoE, while representing a theoretical upper bound for expert diversity, incurs significantly higher costs. While Sparse MoE doubles the parameters of the dense model, its actual training memory is similar due to expert parallelism (Hwang et al., 2023). Our proposed method $M^2$oT, which incorporates loss and its associated router, introduces only a marginal overhead compared to Sparse MoE.

## 6 CONCLUSION

In this paper, we mitigated the compromised trap in agglomerative multi-teacher distillation where student models learn compromised features. Our framework introduces a sparse upcycling Architecture and a specialization-oriented knowledge distillation mechanism. The former designs an efficient sparse student network, while the latter uses a Teacher-Aware Specialization (TAS) Loss to guide Sparse Mixture-of-Experts towards specialized feature learning, grounded in Maximum A Posteriori (MAP) estimation. This enhances expert diversity and improves downstream performance. This beneficial trade-off, coupled with the computational efficiency of our sparse architecture, establishes our method as a state-of-the-art approach.

**Limitations and Future Work.** Despite its efficacy, our framework has limits. The learnable teacher-aware router adds architectural complexity, and the current MAP objective approximation can be improved. Future work will explore more advanced, tractable approximations for the MAP-derived loss. We aim to extend our framework to larger VFM teachers and multimodal distillation settings.

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

## A APPENDIX

We provide our code, more implementation details, additional experiments, and proofs in this section.

### A.1 SOURCE CODE

The anonymous repository for our code is available on `https://anonymous.4open.science/r/M2oT-1CD1/`, which contains the code for training the agglomerative vision foundation models under multiple teachers.

### A.2 THE USE OF LARGE LANGUAGE MODELS

We employ Gemini-2.5 Flash to polish our writing, mainly for the correction of typos and clarity of sentences. Our prompt is as follows: "Please help me polish the sentences in my paper. The sentences are as follows: {TEXT}."

### A.3 IMPLEMENTATION DETAILS

#### A.3.1 DATASETS

We use the popular ImageNet1K and ImageNet21K (Ridnik et al., 2021) as the distillation datasets. We only use the images for feature alignment and discard the labels. The ImageNet100 (Tian et al., 2020) dataset is extracted from ImageNet1K with 100 classes.

#### A.3.2 MODIFIED HIERA ARCHITECTURE

To reduce the latency in both training and inference time, we modified the standard Hiera (Ryali et al., 2023) architecture with: 1) adding window position embeddings, and 2) replacing the window attentions in the first 2 stages with MLP and ReLU attentions.

#### A.3.3 TRAINING HYPERPARAMETERS

We train $M^2$oT-B from scratch and $M^2$oT-L with upcycling Hiera masked autoencoders, the settings of which are shown in both Table 5 and 6. Both models are distilled from 7 teacher models. We train models from scratch in our ablation studies, with the same training settings for all baselines, including epoch, learning rate, batch size and etc. We use A800 GPUs for running our experiments.

#### A.3.4 TEACHER MODELS

To distill sufficient unique features for multiple teachers as much as possible, we select a list of VFMs, including DeiT-III (Touvron et al., 2022), DINOv2 (Oquab et al., 2024), iBOT (Zhou et al., 2022), dBOT-ft (xingbin liu et al., 2024), SigLIP (Zhai et al., 2023), SAM2 (Ravi et al., 2025), and AIMv2 (Fini et al., 2025). We use the base version of DeiT-III, iBOT, dBOT-ft, DINOv2 and the large version of SigLIP, SAM2, AIMv2 as teacher models.

Table 5: Settings of $M^2$oT-B for pretraining on ImageNet1K.

| config | value |
|---|---|
| epoch | 200 |
| learning rate (max.) | 2e-4 |
| learning rate sch. | cosine decay |
| clip gradient | 1.0 |
| warmup epochs | 10 |
| macro batch size | 1024 |
| micro batch size | 128 |
| GPU number | 8 |
| expert parallel | 8 |

Table 6: Settings of $M^2$oT-L for pretraining on ImageNet21K.

| config | value |
|---|---|
| epoch | 20 |
| learning rate (max.) | 1e-4 |
| learning rate sch. | cosine decay |
| warmup epochs | 1 |
| macro batch size | 1024 |
| micro batch size | 64 |
| GPU number | 16 |
| expert parallel | 8 |

## A.4 ADDITIONAL EXPERIMENTS

### A.4.1 COMPARISON OF DIFFERENT TYPES OF MOE

To verify SMoE's effectiveness on agglomeration models, we evaluate various MoE types in Table 7. Soft MoE (Puigcerver et al., 2024) and Sparse MoE with $TOP_1$ active experts show lower accuracies. Sparse MoE with $TOP_2$ experts significantly improves performance, highlighting the benefit of activating more experts. Our method consistently outperforms all other MoE variants. This demonstrates our teacher-aware specialization guidance's superior effectiveness in leveraging sparse MoE for effective multi-teacher knowledge aggregation.

### A.4.2 SPARSE UPCYCLING

To verify the mechanics of sparse upcycling in our framework, we show our results in Table 8. Sparse Upcycling and training from scratch yield similar final performance. However, Sparse Upcycling offers significantly faster convergence by leveraging a pre-trained dense model's parameters as expert initializations. While this provides a strong starting point, a key limitation is that the method is constrained by the dense student model's training hyperparameters.

Table 7: Performance comparison of various MoE configurations on ImageNet100, ADE20K and NYUd datasets.

| | IN100(↑) | ADE20K(↑) | NYUd(↓) |
|---|---|---|---|
| Soft MoE | 92.10 | 35.28 | 0.611 |
| SMoE ($TOP_1$) | 91.68 | 35.02 | 0.636 |
| SMoE ($TOP_2$) | 93.38 | 37.44 | 0.599 |
| $M^2$oT(Ours) | **93.80** | **37.78** | **0.586** |

Table 8: Comparison of initialization scheme of SMoE on model performance.

|  | IN100 | ADE20K |
| --- | --- | --- |
| From Scratch | 93.82 | 37.90 |
| Sparse Upcycling | 93.85 | 37.84 |

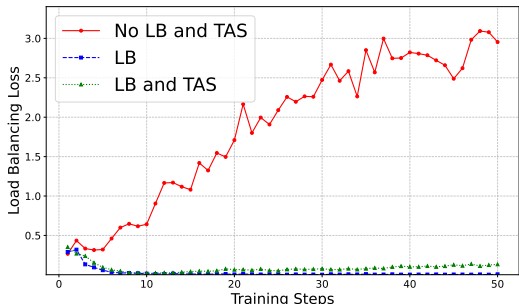

Figure 5: Load-balancing loss curve during training.

### A.4.3 LOAD-BALANCING ANALYSIS

We provide our comparison without adding load-balancing loss, demonstrating that our method achieves load balance despite the tradeoff between load balance and expert diversity, as shown in Figure 5.

## A.5 PROOFS

We provide our proofs for the propositions in our main text. Meanwhile, we also prove the convergence of our proposed loss function.

### A.5.1 NOTATIONS

Let image $x \in \mathcal{I}$ be the input. $f(\cdot)$ denotes the student encoder network. The teacher set $T = \{T_1, ..., T_N\}$ denotes $N$ teacher encoders. The set of SMoE expert $e = \{e_1, ..., e_j\}$ denotes $M$ experts in a network layer.

### A.5.2 COMPROMISED TRAP

To simplify the proof of Proposition 1, we make the following practical assumptions:

- The distance metric is Mean Squared Error (MSE), i.e., $\text{Dist}(f, T_i; x) = \|f(x) - T_i(x)\|^2$.
- Teacher outputs $T_i(x)$ are fixed feature vectors.
- The student encoder $f$ is a single, shared function for all teachers.
- There exist at least two teachers $T_j, T_k$ such that $T_j(x) \neq T_k(x)$ for some input $x$, implying conflicting knowledge.

*proof.* The objective is to find the optimal encoder $f_i^*$ that minimizes the loss for a single teacher $T_i$:

$$f_i^* = \arg \min_f \mathbb{E}_{x \sim D} \left[ \|f(x) - T_i(x)\|^2 \right] \tag{11}$$

Taking the derivative with respect to $f(x)$ and setting it to zero:

$$\nabla_{f(x)} \mathbb{E}_{x \sim D} \left[ \|f(x) - T_i(x)\|^2 \right] = \tag{12}$$

$$\mathbb{E}_{x \sim D} \left[ 2(f(x) - T_i(x)) \right] = 0 \tag{13}$$

Solving for $f_i^*(x)$ yields:

$$f_i^*(x) = T_i(x) \tag{14}$$

The corresponding distillation loss for this optimal encoder is:

$$\mathcal{L}_{\text{dist}}(f_i^*, T_i; x) = \|f_i^*(x) - T_i(x)\|^2 \tag{15}$$
$$= \|T_i(x) - T_i(x)\|^2 = 0 \tag{16}$$

The multi-teacher objective is to find the optimal encoder $\hat{f}$ that minimizes the aggregated loss:

$$\hat{f} = \arg\min_f \mathbb{E}_{x \sim D}\left[\sum_{j=1}^{N} \|f(x) - T_j(x)\|^2\right] \tag{17}$$

Taking the derivative with respect to $f(x)$ and setting it to zero:

$$\nabla_{f(x)} \mathbb{E}_{x \sim D}\left[\sum_{j=1}^{N} \|f(x) - T_j(x)\|^2\right] = \tag{18}$$

$$\mathbb{E}_{x \sim D}\left[2\sum_{j=1}^{N}(f(x) - T_j(x))\right] = 0 \tag{19}$$

Solving for $\hat{f}(x)$ yields:

$$\hat{f}(x) = \frac{1}{N}\sum_{i=1}^{N} T_i(x) \tag{20}$$

Eq. 20 demonstrates that the optimal student output $\hat{f}(x)$ under multi-teacher distillation is the arithmetic mean of all teacher feature outputs. This directly supports our idea of compromised features.

Now we prove that $\mathcal{L}_{\text{dist}}(\hat{f}, T_i) > \mathcal{L}_{\text{dist}}(f_i^*, T_i)$ for some teacher $T_i$.

From the single-teacher derivation, we know:

$$\mathcal{L}_{\text{dist}}(f_i^*, T_i) = 0 \tag{21}$$

From the multi-teacher derivation, the loss with respect to teacher $T_i$ is:

$$\mathcal{L}_{\text{dist}}(\hat{f}, T_i) = \|\hat{f}(x) - T_i(x)\|^2 \tag{22}$$

$$= \left\|\left(\frac{1}{N}\sum_{j=1}^{N} T_i(x)\right) - T_i(x)\right\|^2 \tag{23}$$

$$= \frac{1}{N^2}\left\|\sum_{j\neq i}(T_j(x) - T_i(x))\right\|^2 \tag{24}$$

By our assumption of diverse teacher knowledge ($T_j(x) \neq T_k(x)$), the sum $\sum_{j\neq i}(T_j(x) - T_i(x))$ is a non-zero vector. Therefore:

$$\mathcal{L}_{\text{dist}}(\hat{f}, T_i; x) > 0 \tag{25}$$

This proves that $\mathcal{L}_{\text{dist}}(\hat{f}, T_i) > \mathcal{L}_{\text{dist}}(f_i^*, T_i)$.

Finally, we connect this to the utility function $U$. Let's consider a downstream task $U$ whose optimal performance is highly dependent on the unique features provided by teacher $T_i$. The single-teacher optimal encoder $f_i^*$ perfectly aligns with $T_i$, capturing these essential features and maximizing the utility $U(f_i^*(x))$. In contrast, the multi-teacher solution $\hat{f}(x)$ is an average of all teachers' features and deviates from $T_i$'s unique representation, i.e., $\hat{f}(x) \neq T_i(x)$. This deviation prevents $\hat{f}$ from fully capturing the critical information required by the task $U$, leading to a performance degradation:

$$U(\hat{f}(x)) < U(f_i^*(x)) \tag{26}$$

Thus, the multi-teacher distillation process inevitably results in a compromised trap for tasks that rely on the specialized knowledge of a single teacher.

### A.5.3 TEACHER-AWARE SPECIALIZATION LOSS AS REGULARIZER

As previously shown, under assumptions of MSE loss and a shared encoder, the optimal solution $f_1^*$ is the mean of all teacher features, which represents a compromised solution that sits on a non-optimal point on the Pareto Frontier. The solution $f_1^*$ minimizes the distillation loss, but not necessarily the individual losses or downstream utility. The point on the Pareto front corresponding to $f_1^*$ is often characterized by a high utility trade-off for some specific tasks, meaning its performance is limited on tasks requiring specialized knowledge.

*proof.* The gradient update for $f_1^*$ is the sum of gradients from all individual teachers:

$$\nabla_f \mathcal{L}_{\text{dist}} = \sum_{i=1}^{N} \nabla_f \mathcal{L}_i(f) \tag{27}$$

When teachers possess diverse or conflicting knowledge, their individual gradients $\nabla_f \mathcal{L}_i(f)$ point in different directions. The resulting aggregate gradient $\nabla_f \mathcal{L}_{\text{dist}}$ represents an average direction.

Our proposed $\mathcal{L}_{\text{tas}}$ mitigates this conflict. By guiding individual MoE experts towards a dynamically weighted combination of relevant teachers' features, i.e., $\overline{T}(x, e) = \sum_{i=1}^{N} \pi(x, e, T_i) \cdot T_i(x)$, $\mathcal{L}_{\text{tas}}$ provides a specialized gradient for each expert $e$:

$$\nabla_{\phi(x,e)} \mathcal{L}_{\text{tas}} \propto \nabla_{\phi(x,e)} \|\phi(x, e) - \overline{T}(x, e)\|^2 \tag{28}$$

This specialization-oriented gradient, when combined with the main distillation loss, allows the SMoE experts to move beyond the simple averaging of features and instead learn representations that are tailored to specific teachers.

The term $\mathcal{L}_{\text{tas}}$ encourages expert diversity by pushing each expert to align with a specialized target $\overline{T}(x, e)$, which deviates from the global average target that minimizes $\mathcal{L}_{\text{dist}}$. Thus, there exists:

$$\mathcal{L}_{\text{dist}}(f_2^*) > \mathcal{L}_{\text{dist}}(f_1^*) \tag{29}$$

The regularization from $\mathcal{L}_{\text{tas}}$ enables the model to learn truly specialized features that are highly discriminative for specific downstream tasks. While the averaged solution $f_1^*$ might be mathematically optimal for the aggregate distillation loss, the specialized features of $f_2^*$ provide a much higher utility for downstream applications, proving that:

$$U(f_2^*(x)) > U(f_1^*(x)) \tag{30}$$

Thus, the $\mathcal{L}_{\text{tas}}$ loss within SMoE acts as a regularizer, pushing the model away from a compromised solution toward a more diverse and specialized one that achieves superior performance on downstream tasks.

### A.5.4 CONVERGENCE OF TEACHER-AWARE SPECIALIZATION LOSS FUNCTION

The Teacher-Aware Specialization (TAS) loss for active experts $e$ is formulated as:

$$\mathcal{L}_{\text{tas}}(f; x) = \sum_e \text{Dist}(\phi(x, e), \overline{T}(x, e)) \tag{31}$$

We assume the following standard conditions for the neural network functions and training process:

- The SMoE experts $\phi(\cdot, e)$ and the router network, are differentiable.
- The gradients of the loss functions with respect to the model parameters are well-defined and Lipschitz continuous.
- The distance metric $\text{Dist}(\cdot, \cdot)$ is a differentiable and non-negative function.

*proof.* Let $w$ be be the vector of all trainable parameters of the student model $f$. The total loss $\mathcal{L}_{\text{total}} = \mathcal{L}_{\text{dist}} + \lambda_1 \mathcal{L}_{\text{tas}}$. The optimization process updates these parameters iteratively via gradient descent:

$$w_{t+1} = w_t - \alpha_t \nabla_w \mathcal{L}_{\text{total}} \tag{32}$$

where $\alpha_t$ is the learning rate in the step $t$.

The gradient of the total loss is given by the sum of the gradients of its components:

$$\nabla_w \mathcal{L}_{\text{total}} = \nabla_w \mathcal{L}_{\text{dist}} + \lambda_1 \nabla_w \mathcal{L}_{\text{tas}} \tag{33}$$

where $\nabla_w \mathcal{L}_{\text{dist}}$ is well-defined and can be computed via backpropagation.

For the differentiability of $\mathcal{L}_{\text{tas}}$, it is a sum of differentiable MSE losses over differentiable expert outputs and a differentiable dynamic target. The target $\overline{T}(x, e)$ is a weighted sum of fixed teacher outputs, where the weights $\pi$ come from a differentiable router. Thus, its gradient $\mathcal{L}_{\text{tas}}$ is also well-defined and computable via backpropagation.

Under the assumption of Lipschitz continuous gradients, a gradient descent-based algorithm is guaranteed to converge to a stationary point. Formally, this means:

$$\lim_{t \to \infty} \|\nabla_w \mathcal{L}_{\text{total}}(w_t)\|_2 = 0 \tag{34}$$

This guarantees that the optimization process will eventually reach a state where the gradient is zero, which corresponds to a local minimum. Thus, the TAS loss function ensures the convergence of the training process.

