# OpenReview forum: "M$^2$oT: Agglomerative Vision Foundation Models via Sparse Mixture-of-Experts"
_ICLR.cc/2026/Conference — ICLR 2026 Conference Withdrawn Submission_

### Official Review · Reviewer_qEjB · 2025-10-21

**Soundness:** 2
**Presentation:** 2
**Contribution:** 2
**Rating:** 4
**Confidence:** 3

**Summary:**

This paper proposes M²oT, a novel framework based on MoE for multi-teacher distillation in agglomerative vision foundation models, aiming to address the "compromised trap" where student models learn generalized features that overlook the unique contributions of individual teachers. The framework integrates a sparse upcycling architecture to construct an efficient sparse student network and introduces a Teacher-Aware Specialization (TAS) loss, grounded in Maximum A Posteriori (MAP) estimation, to enhance expert diversity and guide each expert to capture teacher-specific specialized features. Extensive experiments across image classification, semantic segmentation, and depth estimation tasks demonstrate the effectiveness of M²oT.

**Strengths:**

1. It makes sense to use different experts to absorb knowledge from different teachers.

2. The expert specialization is a particularly important topic in MoE.

**Weaknesses:**

1. **Limited generalization**. The student network needs to be a MoE model, but it is not general in real-world applications. Thus, the generalization of the proposed method is limited.

2. **Lacked experiments**:  This paper lacks the analysis about expert load.

3. **Typo**:

(1) In Figure 3, is the main difference between Sparse MoE and M²oT that M²oT includes an expert specialization loss?

(2) After Figure 3a, the authors should introduce Figure 3b, rather than Figure 4a and Figure 4b.

Compared to dense models, it is intuitively reasonable that the knowledge conflict issue in multi-teacher distillation is mitigated in MoE models. However, this does not constitute a strong technical contribution. The severity of knowledge conflicts in multi-teacher distillation naturally varies across different student architectures. Research in this area should aim to develop a more general resolution for conflicts, rather than relying on switching the student model architecture (e.g., to MoE). More importantly, real-world applications may impose constraints on the student model (e.g., it must be non-MoE). For the reason, my recommendation is reject now.

**Questions:**

Please refer to Weaknesses.

---

### Official Review · Reviewer_bz2v · 2025-10-31

**Soundness:** 2
**Presentation:** 2
**Contribution:** 2
**Rating:** 2
**Confidence:** 5

**Summary:**

This paper proposes M^2oT, a framework for multi-teacher distillation of vision foundation models, designed to address the issue of compromised trap. Specially, it introduces a Sparse Mixture-of-Experts architecture and a Teacher-Aware Specification Loss to learn features tailored to teachers' specific representations. Experiments on three vision tasks validate the effectiveness of the proposed method.

**Strengths:**

1. The paper provides a theoretical foundation for the proposed method by including a proof of the compromised trap and a convergence analysis of the teacher-aware specialization loss function.
2. M^2oT attains state-of-the-art performance across three benchmarks with two different backbones, highlighting the strong performance.

**Weaknesses:**

1. A major concern is the lack of discussion and comparison with the closely related work SAK [1], which also addresses multi-teacher distillation of vision foundation models by leveraging the representation biases to combine different teachers' knowledge. However, it is neither discussed nor included in experimental comparisons, as SAK is open-source. Incorporating this work would lead to a fairer and more complete evaluation.

[1] Yuxiang Lu, Shengcao Cao, and Yu-Xiong Wang. Swiss army knife: Synergizing biases in knowledge from vision foundation models for multi-task learning. In The Thirteenth International Conference on Learning Representations, 2025.

2. Proposition 2 appears to rely primarily on empirical observations and may not generalize broadly. The evidence provided in Table 1 is limited to a single experiment. It remains unclear whether this phenomenon persists under different experimental setups such as diverse student backbones or different VFM teachers. Moreover, the experimental details are not included, and the NYUd depth estimation task is missing.
3. The figure quality could be improved. The text and label font sizes in Figures 1 and 4 are too small to read.
4. To better assess the generalizability of M^2oT, it would be beneficial to include additional tasks, such as multi-modal learning (as done in RADIO), to further evaluate the model’s capability across diverse domains.
5. Several conceptual and methodological ambiguities (listed in the following questions) raise concerns about the comprehensiveness and rigor of the work.

**Questions:**

1. What are "ideal latent teacher targets" referred to? I do not find any clear definition or explanation in the paper and this phrase does not appear to be a standard or commonly-used term.
2. How to determine the hyperparameter $\lambda_2$？ The authors only report a fixed value of 0.01 in the implementation details. And I am interested to know the joint influence of the two loss weights $\lambda_1$ and $\lambda_2$, as their balance could also affect the distillation performance.
3. Are the same teachers used for M^2oT-B and M^2oT-L? The paper mentions 7 teachers overall, but only DINOv2 and SAM2 are listed in the performance comparison of Table 2. Can the student model consistently outperform all teachers? Furthermore, could the observed performance gains over prior baselines be partly due to the inclusion of additional teacher models?
4. In the ablation study on scaling the number of teachers (Figure 3a), how are the teachers selected when the number of teachers is set to 1 or 3? Does the choice or combination of teachers affect the distillation performance?
5. In Table 4, is the model used for computation M^2oT-B? I assume the "Dense" baseline is a plain ViT-B backbone. But why the number of parameters reported in Table 2 (86M) differs from the 171M shown in Table 4?

---

### Official Review · Reviewer_vC4q · 2025-10-31

**Soundness:** 2
**Presentation:** 2
**Contribution:** 2
**Rating:** 4
**Confidence:** 4

**Summary:**

This paper addresses the "compromised trap" in multi-teacher distillation, where student models learn a weak average of teacher features. It proposes $M^{2}oT$, a framework using a Sparse Mixture-of-Experts (SMoE) architecture and a novel Teacher-Aware Specialization (TAS) loss. This TAS loss guides SMoE experts to learn specialized features from each teacher. This approach avoids the compromised average and achieves improved performance on diverse vision tasks.

**Strengths:**

1. The paper studies a significant limitation in existing multi-teacher distillation methods, which it terms the "compromised trap". This trap describes how student models learn a generalized, "compromised" average of teacher features, losing the specialized, unique knowledge from individual teachers.

2. The paper proposes the use of Sparse Mixture-of-Experts (SMoE) for agglomerative vision models. This architecture is a natural fit for the problem, providing the modularity and capacity to capture specialized knowledge from different teachers without a proportional increase in computational cost.

**Weaknesses:**

1. Baseline comparisons: The empirical comparisons in Table 2 are not well-controlled. At the base scale, $M^{2}oT$-B significantly underperforms on ADE20K against RADIOv2.5-B (42.50 vs. 48.94 mIoU). At the large scale, the comparison is not fair as $M^{2}oT$-L uses a modern Hiera-Large architecture, while baselines use ViT-Large. Compared to ViT-Large, Hiera-Large is known to generalize well to segmentation tasks such as ADE20K (see SAM2) and can potentially provide a better initialization. To isolate the benefits of the proposed $M^{2}oT$ framework, a controlled experiment using a ViT-Large backbone for $M^{2}oT$-L is needed.

2. Distillation data scale: The work uses ImageNet1K for base models and ImageNet21K for large models, which differ substantially in scale and diversity. The paper lacks an ablation study on how this data scale impacts the severity of the "compromised trap" or the effectiveness of the proposed Teacher-Aware Specialization (TAS) loss. An ablation training the same $M^{2}oT$ architecture on both IN1K and IN21K would provide valuable insight into the method's scalability and its interaction with the data source.

3. Overstated theoretical contribution: The theoretical formalization, particularly Proposition 1 (Compromised Trap) and its proof in Appendix A.5.2, appears to be an unnecessarily complex presentation of a well-understood concept: an optimal solution for an aggregated objective (multi-teacher) is not optimal for individual objectives (single-teacher). The proof that the optimal solution is the arithmetic mean is a standard result and adds little new insight. This section could be significantly simplified, as the core intuition is already well-conveyed by Figure 2, which would improve the paper's readability.

**Questions:**

Please see the Weaknesses listed above.

---

### Official Review · Reviewer_thCs · 2025-11-01

**Soundness:** 2
**Presentation:** 2
**Contribution:** 2
**Rating:** 4
**Confidence:** 2

**Summary:**

The paper addresses the problem of multi-teacher knowledge distillation for building agglomerative vision foundation models that integrate knowledge from several pre-trained visual encoders. Existing multi-teacher approaches (e.g., AM-RADIO, UNIC, RADIOv2.5) tend to produce “compromised” features because a single shared backbone must align with all teachers simultaneously, leading to diluted or averaged representations.
To overcome this limitation, the authors propose $M^2oT$ (Multi-Teacher via Sparse Mixture-of-Experts), a framework that introduces Sparse Mixture-of-Experts (SMoE) layers into the student network. Each expert is encouraged to specialize on a subset of teachers through a Teacher-Aware Specialization (TAS) loss, derived from a geometric-median / MAP formulation, which guides experts toward diverse, teacher-specific representations. A standard distillation loss aligns overall teacher and student features, and a load-balancing loss ensures that experts are evenly utilized. The total objective combines these three terms (Eq. 10).
Experiments are conducted on ImageNet-1K classification, ADE20K semantic segmentation, and NYU Depth v2 estimation, comparing against several recent multi-teacher and agglomerative models (SAM-CLIP, Theia, UNIC, RADIOv2.5).
The proposed method shows improved ImageNet accuracy and depth RMSE, competitive segmentation results, and minimal computational overhead.
Ablations explore the influence of the TAS coefficient λ₁ and provide qualitative visualizations (t-SNE, expert–teacher heatmaps) illustrating expert specialization.

**Strengths:**

The paper tackles an important and timely question: how to combine the knowledge of multiple large-scale vision foundation models into a single unified representation without losing their individual strengths.

In terms of originality, the paper proposes an interesting hybrid design that combines multi-teacher feature distillation with a Sparse Mixture-of-Experts (SMoE) architecture. Although the idea of MoE-based distillation has appeared in related work, the authors’ formulation introduces a Teacher-Aware Specialization (TAS) loss motivated by a geometric-median / MAP interpretation. This regularizer is a creative attempt to enforce expert diversity and teacher-specific specialization within the student model, thereby addressing the “compromised trap” of prior approaches.

Regarding quality, the experiments cover three representative computer vision benchmarks: ImageNet classification, ADE20K segmentation, and NYU Depth estimation. The experiments show that the proposed method is competitive or superior on several tasks.
The empirical evaluation includes two model scales (ViT-Base, Hiera-Large) and demonstrates that the additional SMoE and TAS components introduce minimal computational overhead.
The qualitative analyses (t-SNE plots and expert–teacher heatmaps) offer intuitive evidence of expert specialization, complementing the quantitative results.

In terms of significance, the paper contributes to the growing interest in scalable and modular architectures that integrate heterogeneous foundation models. The idea of using routing-based specialization to balance multiple teacher signals could influence future research on cross-model aggregation and multi-source distillation.

Finally, the overall clarity of motivation and the structure of the proposed loss functions (distillation, specialization, load-balancing) are conceptually sound, even though several technical details are under-explained.
The work presents a promising direction for building unified foundation models through modular, expert-based distillation mechanisms.

**Weaknesses:**

The paper’s central idea is interesting, yet several aspects of the methodology, experimental design, and exposition remain unclear or incomplete, limiting reproducibility and confidence in the claims.

1. Method explanation and clarity.
The description of the proposed framework is difficult to follow. Figure 1 is visually dense but does not clearly explain the data flow between teachers, router, and experts.
The relationship between teachers and experts is only implicit, readers must infer that each expert specializes on one or more teachers through the teacher-aware router, but this is not explained well in the apepr.

2.Teacher-model inconsistency.
Section 5.1 claims that seven Vision Foundation Models (DeiT-III, DINOv2, iBOT, dBOT-ft, SigLIP, SAM2, AIMv2) serve as teachers, but Table 2 lists only DINOv2 and SAM2 under “Teacher Models.” It is unclear whether all seven teachers are used simultaneously, in subsets, or only as baselines. The paper should specify the exact teacher configuration used for each experiment and justify these choices.

3.Missing ablations of the loss components.
The total loss (Eq. 10) combines three objectives (distillation, teacher-aware specialization, and load balancing) yet Section 5.3 does not report results when removing each component. There are no quantitative comparisons for models trained without the TAS loss, without the load-balancing loss, or with both removed. Such ablations are necessary to isolate the contribution of each component and would greatly improve the paper’s empirical soundness.

4.Mixed experimental evidence.
The empirical results at the base scale do not consistently support the paper’s claim of overall superiority.
In Table 2, RADIOv2.5-B (81.96 / 48.94 / 0.498) outperforms M²oT-B (84.28 / 42.50 / 0.479) on segmentation by 6.4 mIoU, despite comparable capacity.
While M²oT shows gains in classification and depth estimation, the global cross-task balance favors RADIOv2.5-B.
Because the method is proposed as a unified agglomerative VFM, evaluation should consider aggregated or normalized multi-task metrics rather than isolated ones.

5. Equation (9) and loss scaling ambiguity.
The load-balancing loss introduces a constant M (number of experts) as a multiplicative factor, yet Eq. (10) multiplies the same term by
$λ_2$. This yields an effective coefficient $M \times \lambda_$ causing the regularization strength to grow linearly with the number of experts.The purpose of this scaling is never justified. The authors should clarify whether this follows Fedus et al. (2022) for normalization, or if they tuned $\lambda_2$ to compensate.

**Questions:**

See each point in Weaknesses.

---

### Note · Authors · 2025-12-02

**Comment:**

We thank the reviewers for their time and constructive comments. After careful consideration of the reviewer feedback，we believe the paper requires more substantial revisions and additional experimental work to address the major concerns. We intend to use this valuable feedback to significantly improve the work.

**Withdrawal Confirmation:**

I have read and agree with the venue's withdrawal policy on behalf of myself and my co-authors.